# A Novel Integration of Face-Recognition Algorithms with a Soft Voting Scheme for Efficiently Tracking Missing Person in Challenging Large-Gathering Scenarios

**DOI:** 10.3390/s22031153

**Published:** 2022-02-03

**Authors:** Adnan Nadeem, Muhammad Ashraf, Kashif Rizwan, Nauman Qadeer, Ali AlZahrani, Amir Mehmood, Qammer H. Abbasi

**Affiliations:** 1Faculty of Computer and Information System, Islamic University of Madinah, Madinah 42351, Saudi Arabia; a.alzahrani@iu.edu.sa; 2Department of Computer Science, Federal Urdu University of Arts, Science & Technology, Islamabad 45570, Pakistan; m.ashraf@fuuast.edu.pk (M.A.); kashifrizwan@fuuast.edu.pk (K.R.); nauman.qadeer@fuuast.edu.pk (N.Q.); 3Department of Software Engineering, Faculty of Engineering, Science, Technology and Management, Ziauddin University, Karachi 74700, Pakistan; amir.mehmood@zu.edu.pk; 4James Watt School of Engineering, University of Glasgow, Glasgow G12 8QQ, UK; qammer.abbasi@glasgow.ac.uk

**Keywords:** tracking missing persons, large-crowd gatherings, integration of face-recognition algorithms, soft voting scheme

## Abstract

The probability of losing vulnerable companions, such as children or older ones, in large gatherings is high, and their tracking is challenging. We proposed a novel integration of face-recognition algorithms with a soft voting scheme, which was applied, on low-resolution cropped images of detected faces, in order to locate missing persons in a challenging large-crowd gathering. We considered the large-crowd gathering scenarios at Al Nabvi mosque Madinah. It is a highly uncontrolled environment with a low-resolution-images data set gathered from moving cameras. The proposed model first performs real-time face-detection from camera-captured images, and then it uses the missing person’s profile face image and applies well-known face-recognition algorithms for personal identification, and their predictions are further combined to obtain more mature prediction. The presence of a missing person is determined by a small set of consecutive frames. The novelty of this work lies in using several recognition algorithms in parallel and combining their predictions by a unique soft-voting scheme, which in return not only provides a mature prediction with spatio-temporal values but also mitigates the false results of individual recognition algorithms. The experimental results of our model showed reasonably good accuracy of missing person’s identification in an extremely challenging large-gathering scenario.

## 1. Introduction

Several events are held each year, around the world, where a huge crowd gathers for some purpose like entertainment or performing religious rituals. It is common practice that some people, especially children and older persons, get separated from their companions in such large-gathering scenarios. Tracking such missing persons efficiently is still a large research problem. The work in hand is an attempt to improve the efficiency of tracking missing persons in large-gathering scenarios of Al Nabvi mosque, Madinah, Kingdom of Saudi Arabia (KSA), where millions of Muslim pilgrims gather every year to perform religious activities.

Person identification, by applying facial-detection and -recognition algorithms efficiently, becomes extremely challenging in large-crowd-gathering scenarios. Major challenges are uncontrolled environments, densely populated regions, and varying quality of facial images (varying from optimal to severely degraded images). The quality of face images is usually degraded, in such a scenario, due to several unavoidable factors including varying image sizes (from very small to medium sizes), low resolution, improper illumination, subjects’ pose variation, the camera’s movement, and the varying distances among persons and cameras. Such challenges made our prepared dataset ideal for evaluating the face-recognition problem in a fully unconstrained environment. Figure 1 shows some sample images from our data set reflecting the complexity of the problem.

The contribution of the proposed work is novel in three aspects. First, it establishes a dataset of images in highly challenging large-gathering scenarios using existing infrastructure available in Al Nabvi mosque. Second, it proposed an integration of five existing face-recognition algorithms with a soft voting scheme for efficient tracking of the missing persons, which was performed for the first time to the best of our knowledge. Third, it achieved implementation results with reasonably good accuracy.

The department of “lost and found” in Al Nabvi mosque is currently working in a manual manner, and the main purpose of this work was to facilitate this department in finding missing persons efficiently through an automated system. This work proposes a geofence markers scheme coupled with twenty cameras installed in the Al Nabvi Mosque. We prepared a dataset of images obtained by those installed cameras. Each image in the dataset was also augmented with spatio-temporal information conveying the geographical location of the capturing camera and the captured time. The proposed system fetches the reported missing-person profile image for tracking in real-time video images. After real-time face detection, using Viola–Jones and the cascading AdaBoost algorithm, we initially applied five well-known face recognition algorithms (PCA, DCT, LBP, LGBPHS, and ASR+) individually to check their performance in challenging scenarios of large-crowd gatherings. However, the results were not satisfactory. Therefore, to optimize the accuracy, we then proposed integration of these algorithms in parallel, and then combined their results with a soft voting scheme for more mature prediction. Finally, the person identification was further improved with spatial and temporal information of missing persons. Although the proposed model was implemented in the Al Nabvi mosque scenarios, the proposed model is general and can be applied to efficiently track missing persons in such large-gathering scenarios.

This article is organized as follows. Section 2 presents the related work in this field and also highlight research gaps. Section 3 describes our proposed model and the spatial distribution of the Al Nabvi mosque along with workflow diagrams, algorithms, and technical details of the proposed model. Section 4 elaborates the implementation workflow of the model and its deployment along with the experimental results. Finally, in Section 5, we conclude our work and highlight some future directions.

## 2. Related Work

We now briefly present our review and gap analysis of the related work. Face recognition in crowded areas with poor-quality photos is a challenging task. Even though the study topic of facial recognition has seen considerable improvement in terms of accuracy and reliability in recent years, the efficacy of findings in confined situations remains a priority. The success in the aforementioned arena is mostly due to the availability of deep neural networks and large amounts of computer power. Nonetheless, the inclusion of large training and evaluation datasets was significant. Among these datasets are Labeled Faces in the Wild (LFW) (13,233 photos) [1], PubFig (58,797 images) [2], FERET (14,126 images) [3], CelebA (200 k images, 10,177 IDs) [4], and YouTube Faces DB (1595 identities) [5]. These datasets include photos of faces in a variety of lighting situations, emotions, races, and positions. Although significant progress has been made in recent years in terms of face-identification recognition accuracy, reconstructing a picture from a low-resolution one remains a difficulty [6]. There is presently no large-scale face dataset that contains many picture sequences. Numerous modern super-resolution techniques are generative in nature and so cannot be employed for identification purposes due to their excessive creativity. Over the last several years, low-quality face recognition (LQFR) has garnered more interest. There are several applications for devices capable of LQFR in real-world conditions where capturing high-resolution or high-quality pictures is difficult or impossible. Video surveillance is one of the important application sectors for LQFR systems. As the number of security cameras rises (particularly in metropolitan areas), the footage they record will need automated processing. However, such films are often shot with enormous standoffs, in difficult lighting circumstances, and from a variety of perspectives. The faces in these photographs are often tiny. Previously published research on this subject has made use of methods such as super-resolution processing, deblurring, and understanding the link between various resolution domains [7].

Jureviius et al. provided a piece of study pertaining to sporting activities. Additionally, it demonstrated that dealing with imagery from a moving source in order to recognize and compare faces in a crowd to an existing face database is a hard scientific issue that demands a sophisticated solution. The purpose of this article was to discuss real-time facial recognition with two algorithms in crowds utilizing unmanned aerial vehicles [8].

Kramer et al. developed the Choke-point Search Test to replicate a real-time search for a target and subsequently found that when three recent photographs of the target were presented during searches, but not three social-media images, performance on the test increased. Taken together, our findings illustrate the inherent problems associated with real-time face recognition, which has significant implications for security professionals who do this work on a regular basis [9].

The authors claimed accuracy on unconstrained faces in still pictures and videos using Amazon Mechanical Turk [10], which is a crowd-sourcing marketplace. They analyzed the accuracy of humans from two different nations (the United States and India) and discovered that people from the United States are more accurate, probably because they are more acquainted with the faces of popular personalities in the YouTube Faces database. A combination of human recognition with a commercially available face-matcher enhances performance above human recognition alone.

Adjabi et al. in [11] provided a study with a specific emphasis on the deep learning technique since it reflects the current state of the art in this sector. Unresolved challenges in face recognition are addressed, and prospective routes for study in the field are presented, in order to serve as a point of reference for subjects deserving of investigation. However, 3D-face-recognition databases outperform 2D databases in terms of accuracy [11].

Xu et al. tested three distinct situations of facial recognition using security cameras. There are several elements that affect the performance of facial-recognition software, that is, the name, the camera type, the distance between the item and the camera, and the resolution of the facial photos collected [12].

As described earlier, the presented work dealt with the scenario of identifying missing persons in an unconstrained environment with low-resolution image data in a crowd. The performance of recognizing and identifying a person degrades in unconstrained environments. Some of the researchers have worked on this problem. We review a few of them as shown in Figure 2, namely [13,14], which worked on the identification of a person in crowded environments with low-resolution images as the source of input. Some other works are [15,16,17,18,19,20,21,22,23]. However, Table 1 depicts the details of the environment constraints, the tools and techniques used, the methodology, the dataset and performance used by various researchers.

Hence, we found the gap in identification of persons under the unconstrained environment having a crowd with low-resolution images and used five state-of-the-art, well-known, and well-explored algorithms to cater the above-said problem; we found a significant improvement in the efficiency of the results as compared to other work as shown in [8], which used two algorithms.

## 3. Proposed Methodology

We present our proposed methodology by first illustrating the coverage of Al Nabvi mosque with twenty cameras with geofences. Then, we present our proposed model in details.

### 3.1. Cameras Setup and Spatial Distribution of Al-Nabvi Mosque

We propose geofences marked and coupled with 20 (twenty) cameras in Al-Nabvi mosque, Madinah, KSA to capture spatial and temporal data for the proposed experiment design as shown in Figure 3. Eight fences in number are in place for the spatial features integrated with temporal artifacts of reported lost or found persons. In our scenario, fences are marked with respect to the easy and convenient on-ground places (i.e., distinct parking and distinct washrooms) across the courtyard of Al-Nabvi mosque. Each fence is required to be swept and covered with a significant number of surveillance cameras with unique identification and spatial and temporal artifacts. Therefore, each clique of cameras covers one of eight fences to monitor lost persons. Hence, these groups cover each fence and record the targeted persons to be found using spatio-temporal features, necessary to locate the personals inside Al-Nabvi mosque.

### 3.2. Integration of Face-Recognition Algorithms with a Soft Voting Scheme

The method was proposed to improve the efficiency of finding the lost person in Al Nabvi mosque, Madinah. This will facilitate the department of lost and found to automate the current manual work. The method integrates the reporting and finding of lost persons in a single framework, where a complainant registers the missing report through a mobile app and provides the basic profile info of the lost person; then, the system fetches the face images of that person from the central database and extracts the learning features, which are necessary to train the face-recognition algorithms. The video streams of the surveillance cameras are continuously monitored, and Figure 4, Figure 5 and Figure 6 present our proposed workflow to carry out this monitoring. The proposed method examines the video streams at the frame’s level and employs the Viola–Jones algorithm for real-time face detection, which first extracts the feature images into a large sample set and then uses the cascading AdaBoost algorithm as the face detector. The algorithm not only locates the human faces but also applies a bounding box around the human face to localize the entire face region. The video frame with localized face regions is presented in Figure 7. These face regions are cropped and pre-processed to enhance and resize the face images (shown in Figure 8), which are then fed to the face-recognition algorithms for determining personal identification. Each recognition algorithm processes the face image accordingly and then returns a (ID, Score) pair as an output. An ID represents the face-image association to a particular registered person, while a score indicates the face matching rank to that registered person. The further details are presented in Table 2, which states that the prediction with score ≤ 0 indicates no valid match to any registered person. The score color-coding may be used to illustrate the prediction scores at bounding boxes, see, for example, in Figure 9, where a bounding box color presents the prediction score, also mentioned as (ID, Score) pairs in sub-captions.

Individual predictions of the five algorithms are presented in Figure 9. As the result indicates in Figure 9d, an individual-recognition algorithm may go wrong and result in a false prediction with a score > 0; therefore, a unique soft-voting technique is introduced to tackle this issue. The voting technique not only matures the ID prediction but also mitigates the false predictions of individual-recognition algorithms. The proposed voting technique is presented in Figure 5, which takes in the predictions from five algorithms and produces a mature prediction with a score lying in the interval of [0 to 25]. The mature prediction can be ranked, if necessary. The rank details are presented in Table 3. The proposed voting technique is novel in its working; it broadly categories all the input possibilities in two major groups: (1) every algorithm results in the (ID, Score) pair with score ≥ 0; (2) only a few algorithms result in the (ID, Score) pair with score ≥ 0.

In case 1, the proposed scheme asks whether a few algorithms have the same prediction and whether their aggregate score ≥ 5 or not. If yes, the particular ID is associated with that human face, otherwise not, and a tag of NM-1/NM-2 is attached, which indicates no valid match for that human face. Similarly, the scheme in case 2 asks whether two or more algorithms have the same prediction and whether their aggregate score ≥ 6 or not. If yes, the particular ID is associated with that human face, otherwise not, and a tag of NM-1/NM-2 is attached, which indicates no valid match for that human face. The mature prediction results can be seen in Figure 10, where only one face region is identified as ID-142, and the remaining two faces have no valid match to any registered personal. This technique also counts for the predictions with a negative score and tends to minimize the false-negative errors up to a certain extent. It intentionally tries (1) not to miss the reported missing persons by chance and (2) avoids the significant increase in false-positive errors. Both factors are necessary to improve the system performance, and the evaluation results, presented in Figure 11 and Figure 12, depict the consistent performance in this regard. When all of the face regions, detected on a frame, are examined, the system tries to determine their presence tracks in a spatio-temporal context.

The proposed scheme of determining the presence tracks is shown in Figure 6, which takes in the IDs for *m* face regions and sequentially combines them to the respective IDs, predicted on a set of previously examined consecutive video frames. These sequences often form the IDs’ rough presence tracks, which are not suitable for system performance. Therefore, morphological closing and opening operators are applied to smooth the IDs’ presence tracks. Suppose *A* represents an ID rough presence track, and *B* the 1-D morphological filter, then a morphological closing operation for that ID track is defined as:(1)A•B=(A⊖B)⊕B
where ⊖,⊕, and • indicate the erosion, dilation, and closing operators, respectively. The 1-D morphological closing operation smooths the 1’s intervals in the ID presence track, which is necessary to minimize the false-negative error on that track. The morphological opening operation for ID track *A* is defined as:(2)A∘B=(A⊕B)⊖B
where ∘ indicates the morphological opening operator. The 1-D morphological opening operation smooths the 0’s intervals in the ID presence track, which is necessary to minimize the false-positive error on that track. The minimum false-positive error significantly reduces the false presence of an ID in personal tracking, which is beneficial for system performance. Similarly, the minimum false-negative error certainly avoids the possible disruptions in real-time tracking, which is necessary to improve the method’s performance regarding tracking and finding the reported missing personals. The proposed tracking scheme generally requires a buffer memory to hold the IDs’ record, predicted on a set of previously examined consecutive video frames. An increase in smoothness of the presence tracks also requires a longer 1-D morphological filter, which in turn also increases the memory demands for holding the longer predictions’ records temporarily.

The workflow intermediate results are illustrated in Figure 7, Figure 8, Figure 9, Figure 10, Figure 11 and Figure 12, where Figure 7 presents the human-face detection on a particular frame in the video sequence. Only three faces were detected and localized by the face detector; the remaining highly degraded face regions could not be localized by the Viola–Jones algorithm. Figure 8 presents the three cropped face regions, which were further enhanced and resized to a size of 50 × 50. Every face image was fed to the five recognition algorithms in parallel, which processed it accordingly and returned the identification result for that face. Figure 9 presents the identification result of the individual algorithm for three input faces, where every algorithm identified one of the input faces as ID-142. The remaining two faces had no valid match to any registered personal in data set; however, the algorithm of LGBP wrongly identified said faces as ID-144. The two unidentified face regions were tagged as NM-1/NM-2, where the tag of NM-2 stands for “No match recommended”, and every algorithm assigns this tag to a face only if it finds a very little match for that face region. The tag of NM-1 stands for “No match suggested due to confusion”, and every algorithm assigns this tag to a face if it finds a little match for that face region. All the five identification results for every face region were fed to the soft-voting scheme that produced a more mature identification for those face regions. Figure 10 presents the mature identification results for three input faces, where two face regions have no match to any registered personal, while one face was identified as ID-142. The mature identification results were fed to the proposed tracking scheme that sequentially combined the ID-142 current prediction to the ID-142 predictions, recorded on a set of previously examined consecutive video frames. This sequence produced a rough presence track for ID-142, which was further smoothed to make the stable and consistent prediction for that ID. Figure 11 presents ID-142 rough and smooth tracks, where a dashed-line manual track for ID-142 is compared against the system-generated solid-line tracks. The comparison between a solid-line track and the dashed-line manual track indicates tracking errors, where positive errors indicate the ID’s false presence in video sequence, and negative errors indicate the ID’s false absence in that sequence. The smooth tracks present a better appearance than rough tracks, and certainly reduce the tracking errors in the video sequence. Figure 12 presents the tracking evaluation of ID-142, which is consistent with the tracking errors in smooth tracks. According to the evaluation results, the system shows a good performance for identifying and tracking ID-142.

The pseudocode for the proposed workflow is presented in Algorithm 1, which include all the major steps, to be taken for spatio-temporal tracking. Line 2 creates the face-detector object, while lines 4–14 indicate the continuous monitoring of video streams. Line 4 reads the *t* (current) frame of the video sequence, and line 5 detects the face region in that frame. Line 7 crops the face region, and line 8 resizes that face to a size of 50×50, which also enhances the image quality up to a certain extent. Line 10 is iterated 5 times, every time it feeds the face image to a recognition algorithm, which returns an (ID, Score) pair as an output. Line 12 gets a mature prediction from input (ID, Score) pairs. Line 14 finds the smooth presence tracks, which indicate the presence of a subject in the video sequence.
**Algorithm 1** Proposed tracking workflow1:// create detector object2:FaceDetect←vision.CascadeFaceDetector(MergeThreshold,5);3:**while** 
on 
**do**4:   $fr^{t}\leftarrow camera$// get video frame *t*…5:   $BB\leftarrow step(FaceDetect,fr^{t});$ // face detection…6:   **for** ∀b∈BB **do**7:     $f\leftarrow fr^{t}\left(b\right)$ // crop the face region8:     $f_{er}\leftarrow imresize\left(f,interpolation,\left[50,50\right]\right);$9:     **for** ∀j∈Alogs **do**10:        ${\left(ID,Score\right)}_{j}\leftarrow algo_{j}\left(f_{er}\right);$// for *j*th algorithm11:     **end for**12:     $\left(ID,Score\right)\leftarrow Voting(\left[{\left(ID,Score\right)}_{1},...,{\left(ID,Score\right)}_{5}\right])$13:   **end for**14:   $Tracks\leftarrow Tracking(IDs^{t},IDs^{t-1},IDs^{t-2},IDs^{t-3})$15:   // go for next frame16:**end while**

## 4. Results

This section first describes our dataset and then presents the implementation results of the proposed model in detail.

### 4.1. Dataset

We established a dataset of 188 personals, including children, youngsters, and elderly individuals from existing infrastructure in Al Nabvi mosque. The dataset contains a total of 3474 face images, which were extracted from the videos, recorded by the cameras on-premises, installed in Al-Nabvi mosque. The quality of face images varies from optimal to severely degraded images, which makes the dataset ideal for evaluating the face-recognition problem in a fully unconstrained environment, which in return also provides an opportunity to evaluate the problem of personal identification in huge crowds. The quality of face images was degraded due to several unavoidable factors, including varying image sizes (i.e., from very small- to medium-size images), low resolution, improper illumination, subject pose variation, cameras’ movement, and the varying personal distance from the cameras. Since the dataset was obtained from a fully unconstrained environment, the number of face images for several personals varied from small to very large numbers (i.e., from 2 to 107 face images). The personals that were captured by the cameras for a short period had a lesser number of images, and the personals that were captured by the cameras for a longer period had face images in large numbers. Due to the unconstrained environment, all of the face images were captured in different sizes; therefore, all of the face images were enhanced and resized to a size of 50 × 50, as shown in Figure 13.

All the 188 personals in the dataset were pilgrims who visited the Al Nabvi mosque in Madina. To prove the concept of finding and tracking the missing personals, half of the randomly chosen pilgrims were taken as the registered personals, and the remaining pilgrims were considered as the unregistered personals. Then, a few of the registered personals were taken as the reported missing personals; see, for example, in Figure 14, which contains the sampled face images of a few personals from every category. This distribution of the personals is necessary to conduct the performance analysis in two aspects: (1) face recognition; (2) personal identification and tracking in spatio-temporal context. Identification, tracking and finding evaluations were conducted by considering a highly challenging large-gathering scenario at Al Nabvi mosque.

### 4.2. Face Recognition

Face recognition plays a vital role in bio-metric attendance, personal identification, finding lost personals, and tracking the personals kept under surveillance. In order to conduct the face recognition over this dataset, eight face images of every registered personal were drawn from the dataset and used to train the five algorithms for face recognition. Then, a total of 2348 remaining images were used as test images, where 1974 test images were drawn from registered personals and 374 test images from unregistered personals. First, we employed the five established algorithms for face recognition and evaluated their performance on this dataset. Then, we employed these algorithms in parallel and combined their prediction by soft-voting technique, which produced a more mature prediction for input images. The established face-recognition algorithms we used here are described in the following subsections.

#### 4.2.1. Face Recognition Using Principal Component Analysis (PCA)

The principal components of a collection of face images are actually the sequence of p Eigenvectors, where the ith Eigenvector is the direction of a line that best fits the face data, while being orthogonal to the first i-1 Eigenvectors. Therefore, all the p Eigenvectors constitute an orthogonal basis, where individual Eigenvectors of the face data are linearly uncorrelated. The PCA is actually a process of computing the principal components of data and using them to perform a change in basis on that data, sometimes using only the first few principal components (Eigenvectors) and ignoring the remaining less-important components. Hence, PCA is commonly known for dimensionality reduction due to only projecting the face images to the first few principal components, to obtain the lower-dimensional face ingredients, while preserving as much of the data’s variation as possible. The workflow of the method for training and testing consists of following eight steps: (1) prepare the face vectors, and compute the average face vector; (2) subtract the average face vector from original face vectors; (3) calculate the covariance matrix; (4) calculate the Eigenvector and Eigenvalues for covariance matrix; (5) keep on the first k-Eigenvectors; (6) calculate the feature vectors for training images; (7) read the test face and calculate its feature vector; (8) compare the test-feature vector against the stored-feature vectors and find the face class with the minimum Euclidean distance, which shows more similarity with test face. The method is computationally lesser expensive and showed a good performance on our data set—see, for example, in Figure 15. The method showed a reasonable performance when three images of every registered personal were used for learning; however, the performance improved when the number of images set for learning was increased.

#### 4.2.2. Face Recognition via Discrete Cosine Transform (DCT)

A discrete cosine transform expresses the finite sequence of data points in terms of a sum of cosine functions oscillating at different frequencies. It is a widely used transformation technique in signal processing and data compression. The two-dimensional discrete cosine transform (DCT 2) is generally used in JPEG compression; however, it can be used to extract the discriminating features from face images. When applied to the entire face image, low frequency DCT coefficients are selected, because they carry most of the information regarding personal face appearance. This gives the global facial features for personal identification. When face images have better quality, and when facial features such as the chin, the mouth, the nose, the eyes, and the ears are prominently clear, then DCT can be applied to the pre-localized facial features, in order to extract the more discriminating local features for face appearance. This gives the local facial features for personal identification. As the face images in our dataset have a low resolution, the DCT is applied to face images, in order to extract the global facial features. Here, for making personal identification, first the test image is cropped to get only the face image; then, DCT is applied to get the global facial features, and finally the obtained feature vector is compared against the stored-feature vectors, where the nearest-neighbor discriminant analysis is performed to get the face identification. The method is computationally less expensive and showed a better performance on our data set. Figure 15 shows the performance evaluation for DCT; the method shows a good performance when three images of every registered personal were used for learning; however, the performance improved when the number of images set for learning was increased.

#### 4.2.3. Face Recognition via Local Binary Patterns (LBP)

The LBP operator was originally designed to extract the texture features; it is one of the best performing texture descriptors and has been widely used in various applications of computer vision and pattern recognition. The LBP operator assigns a label to every pixel in an image by thresholding the 3 × 3 neighborhood of that pixel with the center pixel value and considering the result as binary number. Then, a histogram of the labels can be used as a texture descriptor. It is also used to express the texture description of the facial appearance. The overall framework of this method consists of the following four steps: (1) the face image is divided into several local regions, then a 3 × 3 LBP operator is applied to get the texture information for every local region; (2) a histogram is computed from the LBPs of every local region; (3) all the local histograms are concatenated to construct an enhanced texture descriptor; (4) the test face texture descriptor is compared against the stored face descriptors, where a comparison is made by using the weighted Chi square distance. The method is computationally less expensive and showed a good performance on our data set, which can be seen in Figure 15. As compared to the PCA and DCT, the method showed a better performance at all; however, the performance had a slight decline when six face images or fewer were used in learning for every registered personal.

#### 4.2.4. Face Recognition via Local Gabor Binary Pattern Histogram Sequence (LGBPHS)

In order to get a more discriminating texture description, the method combines the LBP operator with the Gabor filter, which is a linear filter in image processing and is generally used for texture representation and discrimination. The Gabor filter finds the existence of specific-frequency content in specific directions in a localized image region. The frequency and orientation representations of Gabor filter are claimed to be similar to those of the human visual system. Therefore, an image analysis with Gabor filters is somehow considered to be similar to human visual perception. The method models a face image as a histogram sequence, obtained by concatenating all the histograms of local Gabor binary patterns. The overall framework for this method consists of the following five steps: (1) The test image is cropped to get the only face image and is normalized to obtain the Gabor Magnitude Pictures (GMPs) with various frequencies and different orientations; (2) each GMP is divided into local sub-regions, then a 3 × 3 LBP operator is applied to get the local Gabor binary patterns (LGBPs); (3) a histogram is computed from the LGBPs of every local region; (4) all the local histograms are concatenated to construct an enhanced LGBPHS texture descriptor; (5) the test-face texture descriptor is compared against the stored-face descriptors, where the nearest neighborhood is exploited to make the final classification. The method is computationally expensive but shows a good performance on our data set, which can be seen in Figure 15. As compared to the PCA and DCT, the method showed better performance at all and had the same accuracy level as the LBP on our dataset. However, the performance of LGBP had a greater decline in accuracy when six images or fewer were used in learning for every registered personal.

#### 4.2.5. Face Recognition via Adaptive Sparse Representations of Random Patches (ASR+)

Unlike the holistic face-recognition approaches, the method only requires a sparse representation of randomly selected patches, taken from the face image, as an input. In the learning stage, the method takes a certain number of randomly selected patches, extracted from gallery (training) images of a personal, and constructs a representative dictionary for that personal. A separate dictionary is constructed for every enrolled personal. In the test stage, a certain number of randomly selected patches are extracted from the query image; then, for every test patch, an adaptive dictionary is built by concatenating the best representative dictionaries from the learning stage. Every test patch is classified by using its sparse representative dictionary, and then a query image is classified by patch voting of pre-classified test patches. As per the author’s claims, the method is more robust and efficient than its precedent sparse representation approaches in three aspects: (1) it incorporates the patch-location information in the learning and test stages; (2) the personal dictionaries are further divided into sub-dictionaries, which avoids the requirements of huge time-consuming dictionaries, and now the pre-learned sub-dictionaries are used to construct a test-patch representative dictionary; and (3) the test patches are selected according to a score value that avoids the selection of patches having lesser discriminative information, such as sunglasses. The method is computationally less expensive and showed a much better performance on our data set, which can be seen in Figure 15. As compared to the PCA, DCT, LBP, and LGBP, the method showed a better performance when six or more images were used in learning for every registered personal; however, the performance showed a significant decline for the number of images lesser than six.

#### 4.2.6. Face Recognition via Integrating the Tested Approaches in Parallel

Here, all the five approaches were combined in parallel, and a query image was fed to all the five approaches simultaneously. Every method classified the query image separately and provided a (ID, Score) pair as an output. All the five pairs were further fed to a soft-voting technique, which then returned mature recognition results. This method showed a better performance than all the five individual approaches, which can be seen in Figure 15 and Figure 16. It showed an improved and stable performance, when six or more images were used in learning for every registered personal.

### 4.3. Personal Identification and Tracking

Personal identification is based on bio-metrics, which generally include iris recognition, fingerprint recognition, and face recognition. Iris recognition and fingerprint recognition both require a user’s deliberate involvement in the process; otherwise, we cannot have the thumb impression or the iris pattern from the user. On the other hand, face recognition does not require a user’s deliberate involvement; therefore, it is used for conducting the personal identification and tracking at the same time. Manual presence tracks were obtained for all the registered personals from the video sequence, where 1’s intervals show the temporal information regarding the personal presence in the video sequence, while the camera ID and its orientation represent the spatial context of the personal presence in the marked geofence. Therefore, spatio-temporal information of a registered personal can be used to track him inside the marked geofence at Al Nabvi mosque, Madina.

The proposed workflow for tracking of the personals is presented in methodology section; here, we present the comparative implementation results in Figure 17, Figure 18 and Figure 19. Figure 17 presents the tracking results of 16 personals in the temporal context, where system-generated solid-line tracks are compared against the dashed-line manual tracks. A comparison between the solid-line track and the respective manual track indicates the tracking errors of a particular ID in video sequence, where positive errors indicate the ID false presence in the video sequence, and negative errors indicate the ID false absence in that sequence. Several IDs’ false presence/absence can be observed in their tracks. According to the tracking results, the proposed method shows a better tracking in the temporal context than individual approaches. Figure 18 presents the tracking evaluation of 94 personals, where some the personals were tracked more accurately than others. According to the evaluation results, the proposed approach tracked the 94 individuals more accurately than the other approaches. The precisions were high, but the recalls were much lower than precisions, which in return lowered the tracking accuracies of the respective personals. The accuracies for 94 individuals ranged from 10% to 96%, which were lowered due to the negative errors in presence tracks. A face detector could not detect the severely degraded face regions in video frames, which caused the significant false-negative errors in the presence tracks. There were several factors that caused the image degradation in the video recordings, which in return reduced the system performance for personal tracking. Figure 19 presents the overall performance for 94 personals, where recalls were much lower than the respective precision, and overall accuracies ranged from 38% to 48%. Again, the proposed approach showed a better performance than individual approaches, and 48% accuracy made the system performance acceptable at the first step. Here, the slight decrease in precision was caused by the recognition algorithms, but the recalls were significantly lowered due to the performance lacking by the face detector, which could not detect the severely degraded face regions in video sequences, recorded under the fully unconstrained environment. The face regions in video frames were severely degraded and could not be localized even by the well-established face-detection algorithm.

## 5. Conclusions and Future Work

Tracking missing persons in large gatherings is still challenging. Therefore, considering the importance of this issue in this study, we presented our research supported by a funded research project. In this work, we made our contribution in three aspects. First, we established a dataset of images in a large-gathering scenario of Al Nabvi mosque using the existing infrastructure. Second, we integrated the established face-recognition algorithms in parallel and proposed a novel soft-voting technique that produces a mature identification from the immature results, predicted by the individual-recognition algorithms. Third, the implementation results of the proposed integration showed efficiency in terms of the optimization of accuracy and the reducing of negative results in a challenging large-gathering scenario at Al Nabvi mosque. In the future, we will perform comprehensive testing of the proposed solution in collaboration with the department of lost and found in Al Nabvi mosque.

## Figures and Tables

**Figure 1 sensors-22-01153-f001:**
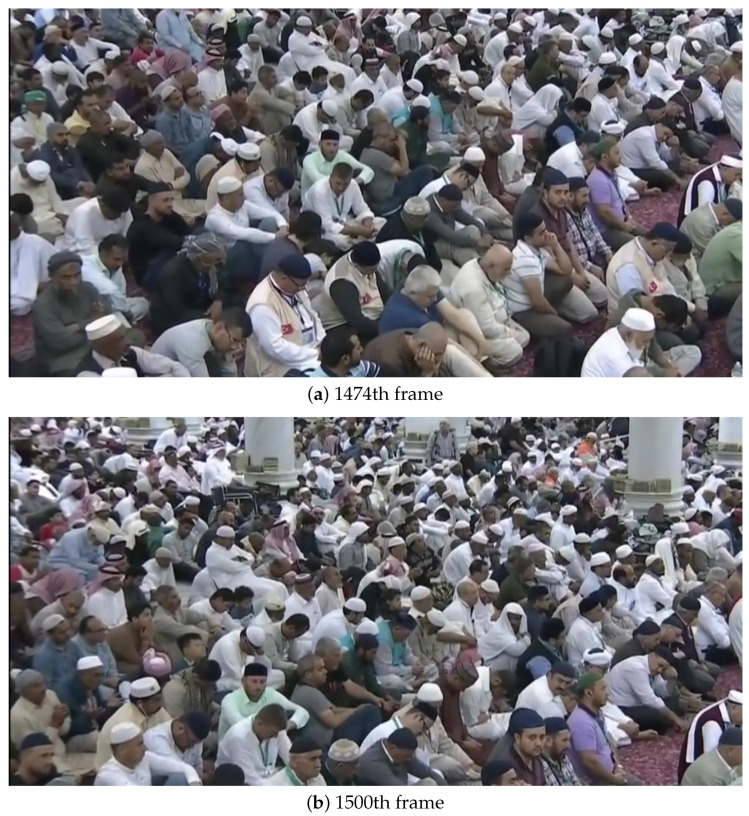
Sampled frames of a video sequence showing the large crowd gathering in Al Nabvi mosque, Madinah (KSA).

**Figure 2 sensors-22-01153-f002:**
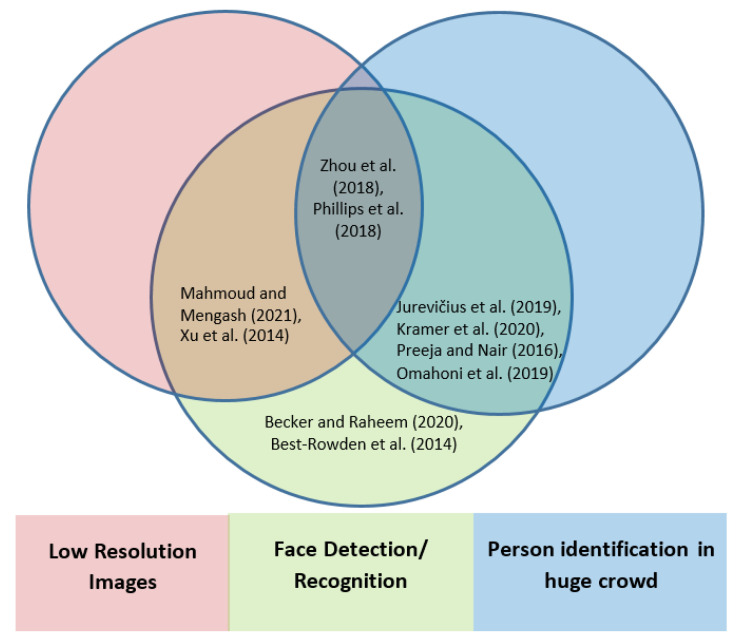
Related work exploration in identifying missing persons in a huge crowd by face detection with low-resolution images [8,9,10,12,13,14,15,16,17,18].

**Figure 3 sensors-22-01153-f003:**
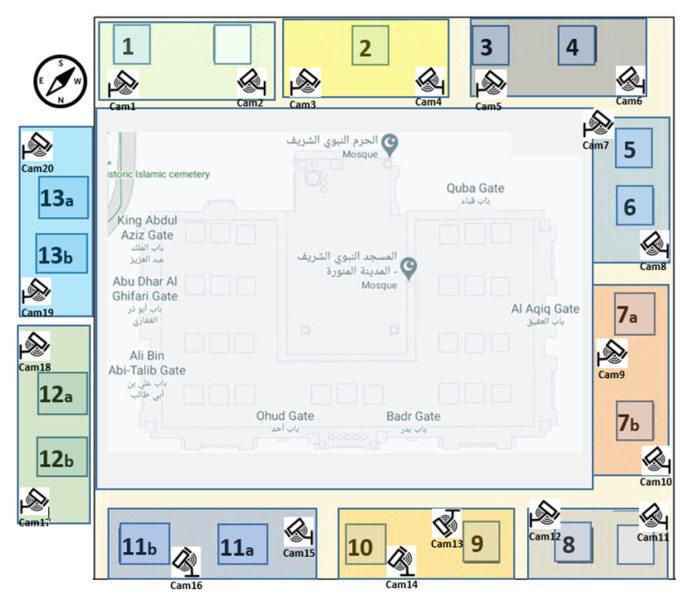
Proposed cameras setups and spatial distribution of Al-Nabvi mosque, Madinah.

**Figure 4 sensors-22-01153-f004:**
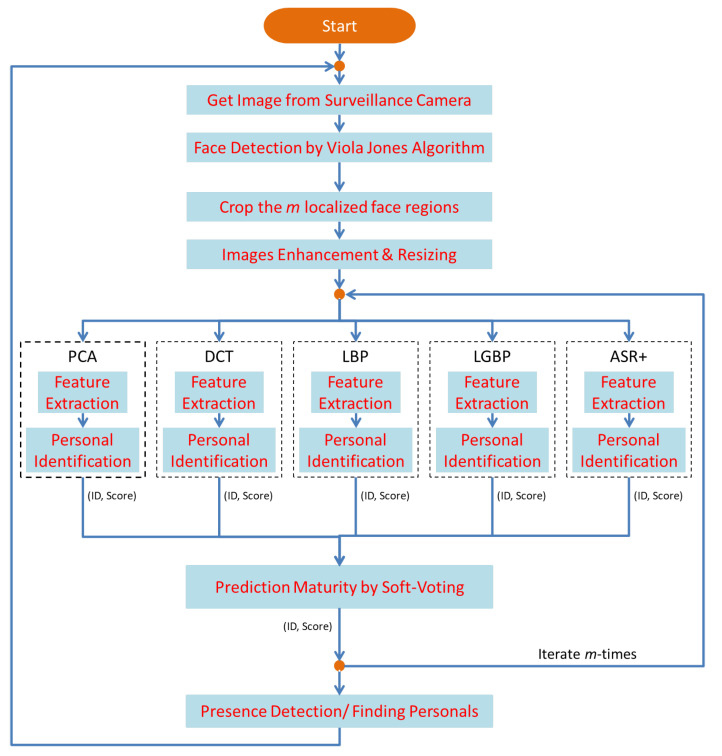
Proposed methodology workflow diagram.

**Figure 5 sensors-22-01153-f005:**
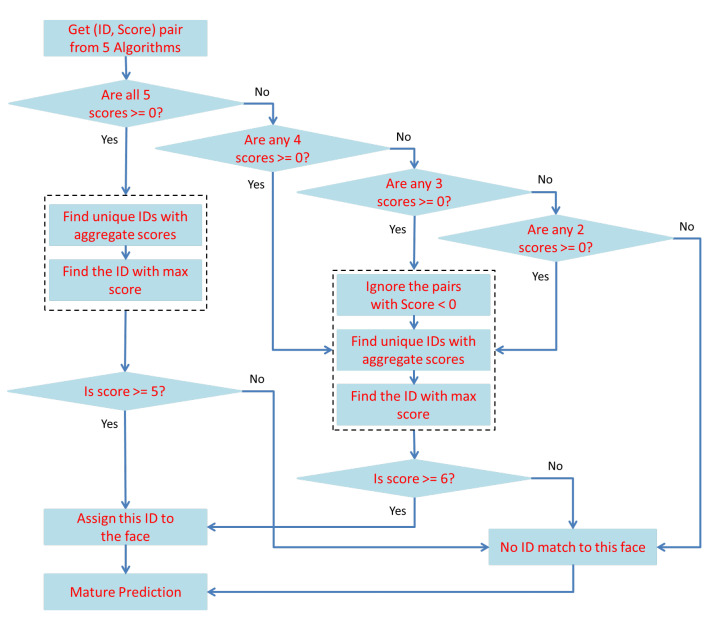
Prediction maturity by soft voting.

**Figure 6 sensors-22-01153-f006:**
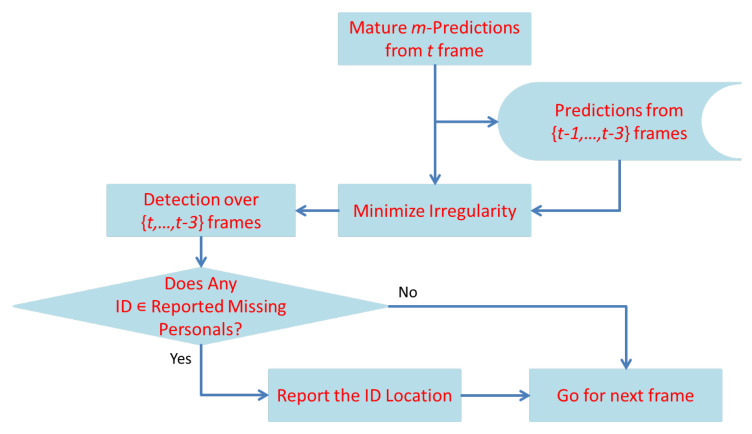
Tracking in spatio-temporal context.

**Figure 7 sensors-22-01153-f007:**
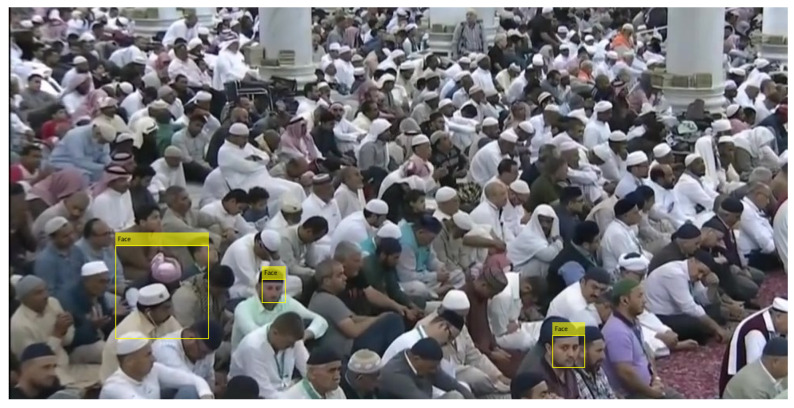
Detected face regions.

**Figure 8 sensors-22-01153-f008:**
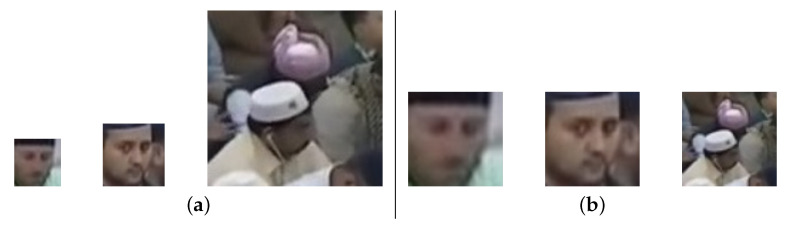
Extracted face regions: (**a**) cropped; (**b**) Enhanced & resized.

**Figure 9 sensors-22-01153-f009:**
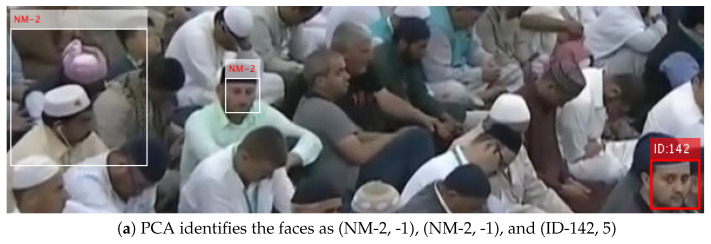
Personal identification by individual algorithms, where (ID, Score) pairs in every sub-caption indicate the face match and its score.

**Figure 10 sensors-22-01153-f010:**
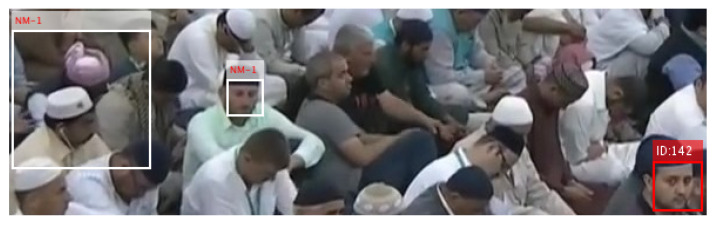
Prediction after maturity by soft-voting.

**Figure 11 sensors-22-01153-f011:**
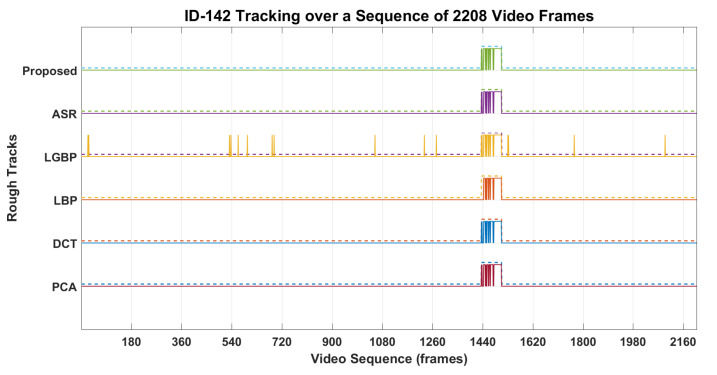
Tracking results of ID-142, where both rough and smooth presence tracks are shown. (The dashed lines represent manual and solid lines the system generated tracks).

**Figure 12 sensors-22-01153-f012:**
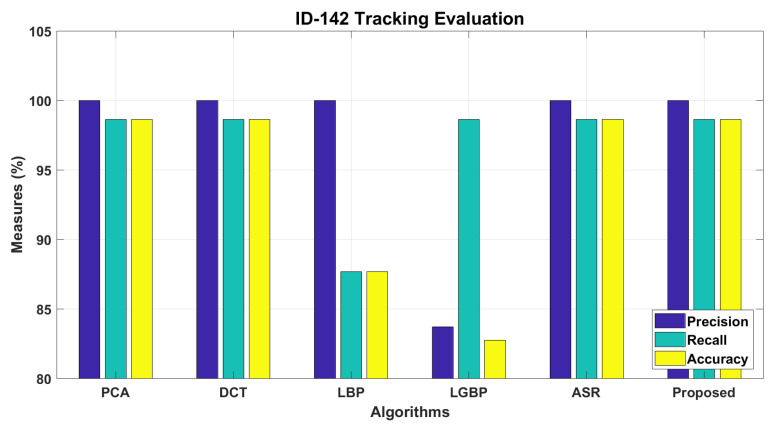
Performance evaluation for identifying and tracking ID-142.

**Figure 13 sensors-22-01153-f013:**
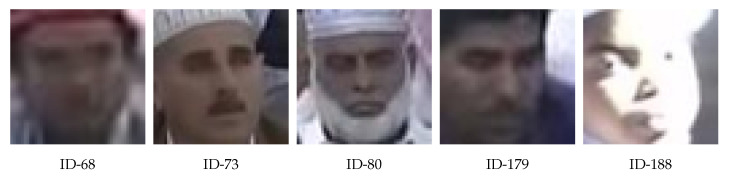
Sampled personal images of size 50 × 50.

**Figure 14 sensors-22-01153-f014:**
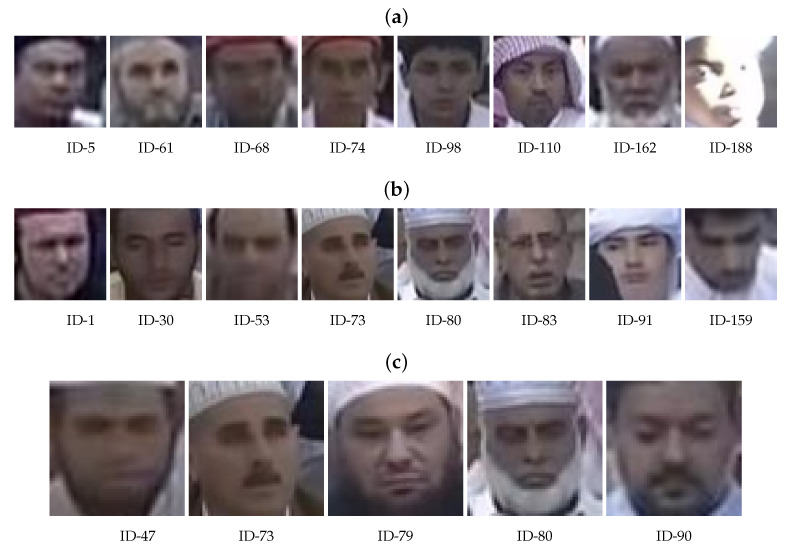
Sampled face images from the dataset, where (**a**) shows un-registered personals, (**b**) registered personals and (**c**) the registered missing personals.

**Figure 15 sensors-22-01153-f015:**
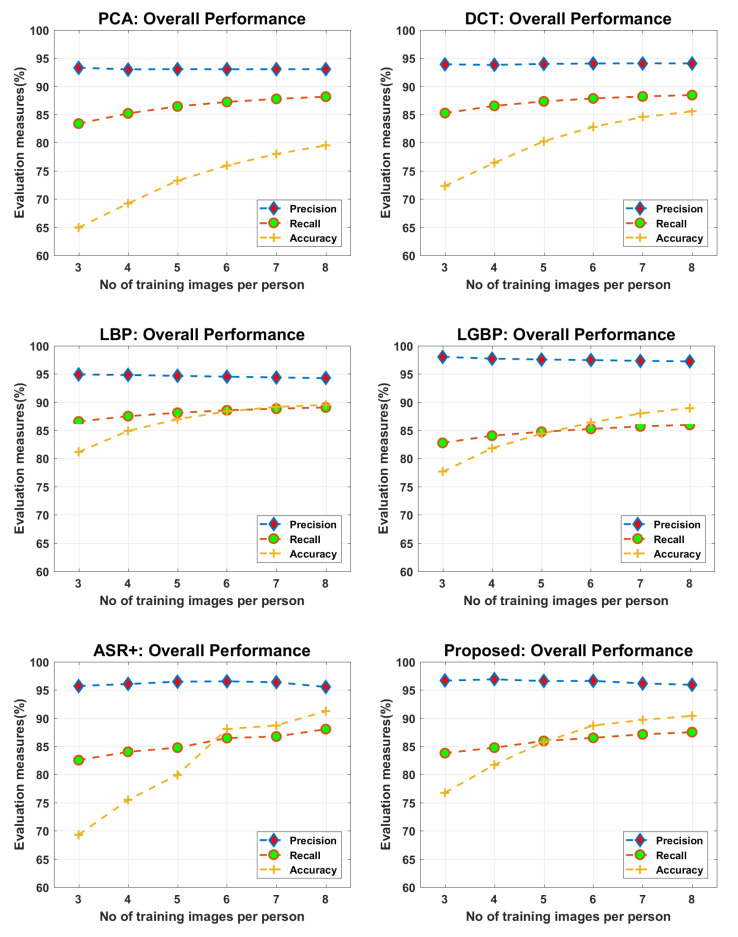
Face-recognition comparative analysis.

**Figure 16 sensors-22-01153-f016:**
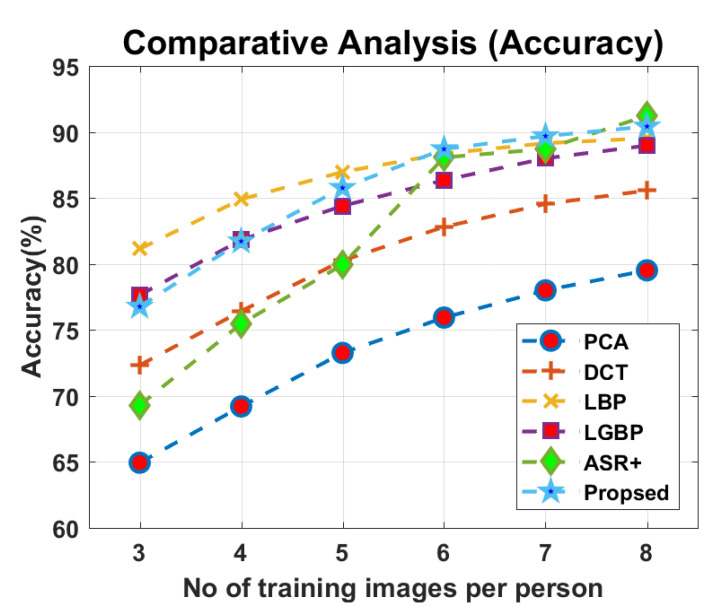
Comparative analysis on accuracy.

**Figure 17 sensors-22-01153-f017:**
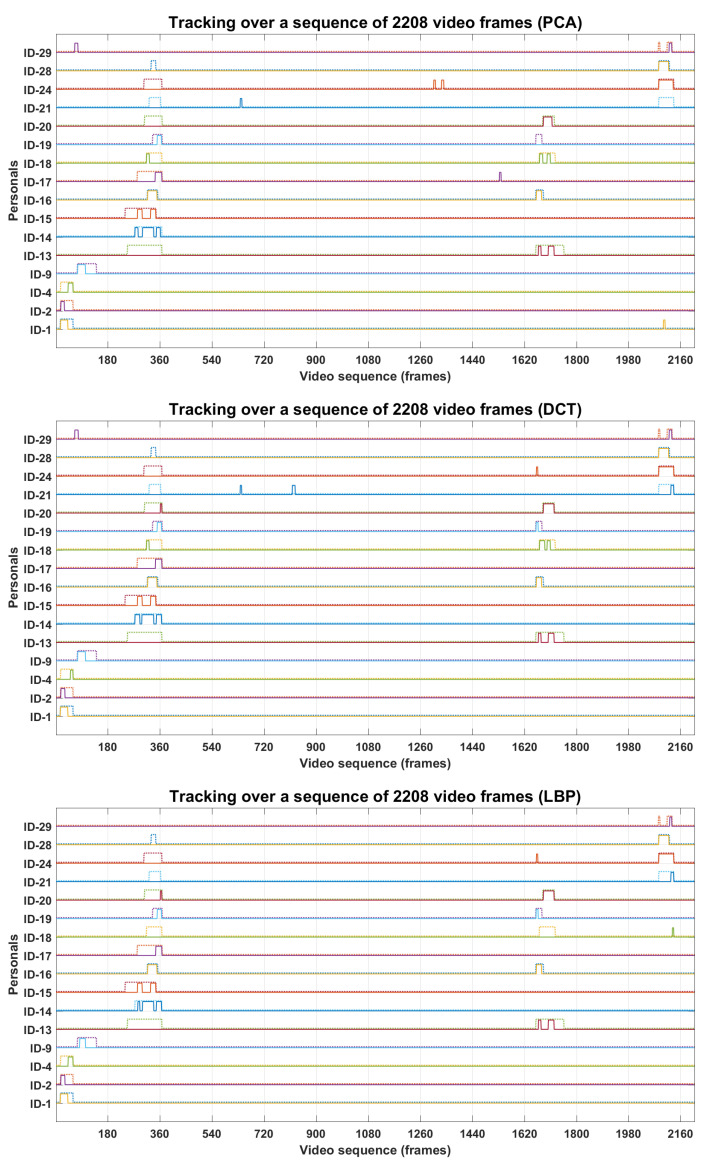
Tracking results of 16 personals.

**Figure 18 sensors-22-01153-f018:**
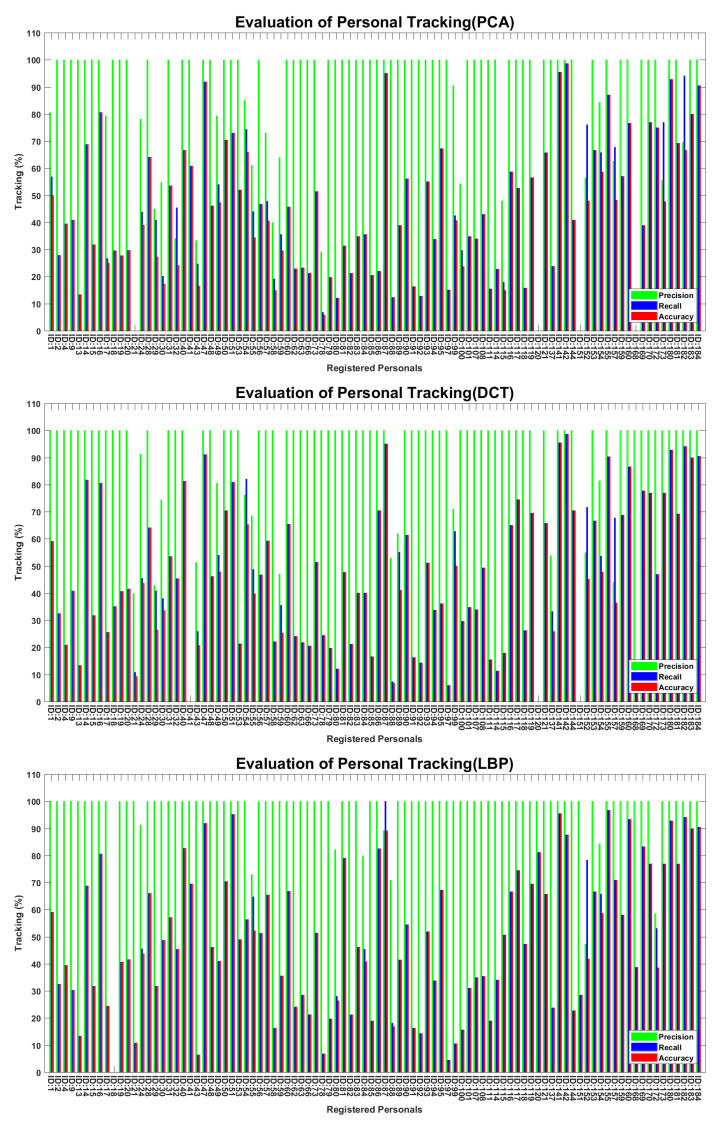
Evaluation of personal tracking in temporal context. Recalls and accuracies were much lower than precisions. Proposed-approach performance was better than individual algorithms.

**Figure 19 sensors-22-01153-f019:**
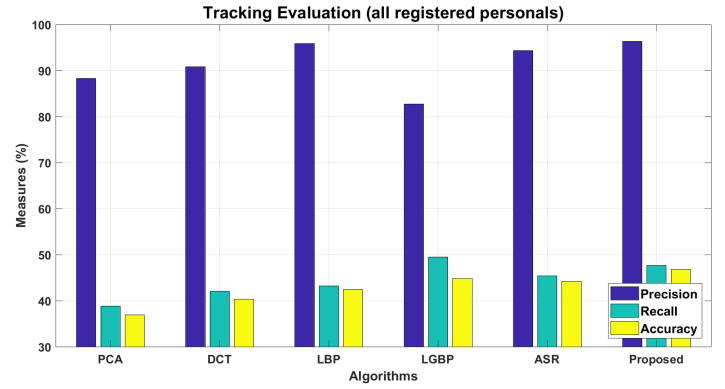
Evaluation summary of temporal tracking for all registered personals.

**Table 1 sensors-22-01153-t001:** Comparison of related work-based selected parameters.

Reference/ System	Input/Environment Constraints			Techniques/Methodology/ Application/Goal/Dataset
	Huge Crowd	Face-Detection (D)/ Recognition (R)	Low Resolution	
[8]	Yes	(R)	NA	• OpenCV • Dlib libraries
[9]	Yes	(R)	NA	• Chokepoint-Search-Test-based simulation in real-time
[10]	Yes	(R)	NA	• Amazon Mechanical Turk
[13]	No	(D)	Yes	• Benchmark dataset FDDB • Hand-crafted and deep-learning-based face detectors (both are not robust)
[14]	Yes	(D)	YES	• DCNN
[15]	Yes	(D)	No	• Occlusion handling • ormalized pixel difference • Crowd density estimation • Real time
[16]	No	(D)	No	• Concealed face detection for masked and unmasked people • Real time • Hybrid non-linear transform model
[18]	Yes	(R)	YES	• Treisman’s visual search methodology
[24]	Yes	(R)	Yes	• Multi-frame face super-resolution model • Dataset available at http://splab.cz/mlfdb/#download (accessed on: 20 December 2021). • Real time

**Table 2 sensors-22-01153-t002:** Prediction ranks table.

Rank	Prediction Score	Description
Rank 5	5	Strongly recommended match to a particular registered person.
Rank 4	4	Recommended match to a particular registered person.
Rank 3	3	Strongly suggested match to a particular registered person.
Rank 2	2	Suggested match to a particular registered person.
Rank 1	1	Weakly suggested match to a particular registered person.
No Rank	0	No match suggested due to confusion (i.e., NM-1).
No Rank	−1, −2	No match recommended (i.e., NM-2).

**Table 3 sensors-22-01153-t003:** Mature prediction ranks table.

Rank	Are All 5 Scores ≥ 0 ? Minimum Aggregate Score	Are Any 4 Scores ≥ 0 ? Minimum Aggregate Score	Are Any 3 Scores ≥ 0 ? Minimum Aggregate Score	Are Any 2 Scores ≥ 0 ? Minimum Aggregate Score
Rank 5	21			
Rank 4	16	18		
Rank 3	11	16	13	
Rank 2	6	11	11	8
Rank 1	5	6	6	6
No Rank	4	5	5	5
No Rank	0–3	0–4	0–4	0–4

## Data Availability

Currently, the data we worked on is not publicly available as yet. However, in future we will make it available for further research.

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
