# Peer review of "A Novel Integration of Face-Recognition Algorithms with a Soft Voting Scheme for Efficiently Tracking Missing Person in Challenging Large-Gathering Scenarios"

_sensors, 2022, doi:10.3390/s22031153_

Round 1
Reviewer 1 Report
The authors have proposed novel integration of face recognition algorithms with soft voting scheme to locate missing person from low resolution cropped images of detected faces from large crowd gathering scenarios. Their proposed method has provided good results as compare to individual implementation of face recognition algorithms in experimented scenarios of Al-Nabvi mosque’s large gatherings. The paper seems technically sound and contributing existing body of knowledge. However, there are few suggestions/observations that need to be implemented for acceptance of this paper. Those suggestions are following:
- There are some places where capital letters are unnecessarily used, extra spaces used and also minor spelling mistakes at few places (e.g. should use ‘department’ instead of ‘Department’ in line 43, ‘Nabvi’ instead of ‘NAbvi’ at line 134, ‘geofences’ instead of ‘Geo fences’ at line 134, similarly authors have used ‘Geofence’ at several places in paper whereas it should be ‘geofence’ instead). The authors need a careful review for such minor mistakes.
- Table 1 (i.e. comparison of related Work) should be in section 2 (i.e. Related Work) instead of section 3.
- Figure 3 should be under section 3.1
- There should be a brief explanation of pseudocode of tracking workflow given in Algorithm-1 at page 10.
- Some explanations also recommended with technical terms used in Figure-9 at page 11 (e.g. what is (NM-1,-2), (ID-142,4) etc. ? )
Overall, it’s nice effort to improve face recognition results in such challenging large gathering scenarios. I recommend it to be accepted for publication after accomodating above suggestions.
Author Response
Please see the attachment for point to point response to the reviewer 1 comment.
Regards

Reviewer 2 Report
This paper provides a voting scheme to combine 5 well-known face recognition algorithms and demonstrates the integrated method performs slightly better. The topic is interesting to find and track missing people from 12 videos for monitoring a large gathering scenario at Al Nabvi mosque. The paper is well written.
However, the technical innovation is low. The soft voting method in Fig. 5 and tracking algorithm are too simple. So, the experimental results show the integrated method is only slightly better than any one of the five applied methods.
Detailed improvement suggestions:
(1) It's better to compact Fig. 17-21 together using maximal half page to do a better comparison.
(2) Figures 22 and 23 cannot sharply show what's the difference between the proposed method and the counterpart methods.
(3) Results in Figure 24 are not good enough. Try to improve your method.
Author Response
Please see the attachment file for response to reviewer 2 comments point to point.

Round 2
Reviewer 2 Report
My comments have been addressed in this version.